Effects of exercise intervention on falls and balance function in older adults: a systematic review and meta-analysis

Yu Haoran
Zhong Jianwei zhongjw04@jxnu.edu.cn
Li Min
Chen Shuainan
College of Physical Education, JiangXi Normal University , Nanchang , China
Khoo Selina
Electronic publication date: 2025 Oct 17
Publication date: 2025
Volume: 13
Electronic Location ID: e20190
Received 2024 Dec 6; Accepted 2025 Sep 15
Copyright: ©2025 Yu et al.
Copyright year: 2025
Copyright holder: Yu et al.
License: This is an open access article distributed under the terms of the Creative Commons Attribution License, which permits unrestricted use, distribution, reproduction and adaptation in any medium and for any purpose provided that it is properly attributed. For attribution, the original author(s), title, publication source (PeerJ) and either DOI or URL of the article must be cited.
License URL: https://creativecommons.org/licenses/by/4.0/

Keywords: Exercise intervention, Older adults, Falls, Balance function, Meta-analysis

Funding: National Social Science Foundation Project 19ZDA353 This study was supported by National Social Science Foundation Project [19ZDA353]. The funders had no role in study design, data collection and analysis, decision to publish, or preparation of the manuscript.

==============================
Objective

To systematically review the effects of an exercise intervention on falls and balance function in older adults (aged > 60 years) without diagnosed diseases.

Methods

PubMed, Web of Science, Embase, Cochrane Library, and CNKI databases were searched for randomized controlled trials about exercise intervention on falls and balance function in older adults. Use Review Manager 5.4 to test the risk bias in the included literature, and use Stata17 for publication bias test, sensitivity analysis, combining effect sizes, forest plots, and subgroup analysis.

Results

A total of 37 randomized controlled trials were included, and meta-analysis showed that after the exercise intervention in the intervention group, there was a significant increase in the Modified Fall Efficacy Scale (MFES) score (g = 1.01, 95% confidence interval (CI) [0.63–1.40], P = 0.00), the number of falls (odds ratio (OR) = 0.32, 95% CI [0.20–0.51], P = 0.00), the Berg Balance Scale (BBS) score (g = 0.92, 95% CI [0.63–1.21], P = 0.00) and Timed Up and Go Test (g = −0.62, 95% CI [−0.80, −0.45], P = 0.00) indices improved better than the control group. Subgroup analysis showed that single exercise time > 30 min, 3 times per week for 12–23 weeks was the better intervention for fall efficacy in older adults, and single exercise time ≤ 30 min, 3 times per week for ≥ 24 weeks was the better intervention for balance function in older adults.

Conclusion

Exercise intervention can enhance fall efficacy, reduce the number of falls, and improve balance function in older adults, and have a certain preventive effect on falls. Single exercise time, exercise frequency and exercise cycle are important factors affecting the effectiveness of exercise intervention. Systematic review registration: https://www.crd.york.ac.uk/PROSPERO/, identifier: CRD42024590937.

Introduction

Falls among older adults constitute a significant global public health concern. World Health Organization (WHO) data indicate that 28%–35% of older adults aged 65 years and older fall each year. The proportion rises to 32–42% among adults aged 80 years and older (Du et al., 2022). In China, falls accounted for 40.88% of major injury-related deaths among adults aged 65 years and older, making falls the leading cause of injury death in this age group (Li, Liu & Ma, 2022). Falls are not only a major cause of injury-related death and disability in older adults, but also an important contributor to their functional impairment, reduced quality of life, and social isolation (Rubenstein & Josephson, 2002). Falls can cause fractures, head injuries, and soft-tissue injuries (Li et al., 2018). In severe cases, falls may be fatal (Lin & Huang, 2016) and impose a substantial burden on families, hospitals, and society (Carty et al., 2015). Older adults who experience falls may develop psychological problems, such as fear of falling. This fear can reduce physical activity, accelerate functional decline, and create a vicious cycle (Zijlstra et al., 2007). The high prevalence of falls in older adults is closely related to the gradual deterioration of their physiological functions. With age, older adults experience a significant decrease in muscle strength and flexibility, a slowing of neural responses, and a weakening of vestibular function and visual perception, all of which combine to affect balance function (Howe et al., 2011). Deterioration in balance function, which encompasses postural control, gait stability, and the ability to adjust to environmental changes, is a major contributing factor to falls in this population (Pua et al., 2017). Older adults with impaired balance fall at higher rates than those with normal balance. They are more likely to lose stability in complex or unexpected situations, which increases their risk of falling (Wang et al., 2019). Consequently, improving balance function has become a central target in fall prevention strategies for older adults.

In recent years, exercise interventions, as non-pharmacological approaches, have received considerable attention for preventing falls and improving balance in older adults (Freiberger et al., 2012). Exercise enhances neuromuscular coordination, improves vestibular function, and helps older adults maintain postural control, thereby reducing fall risk in dynamic or unstable environments (Muir et al., 2010). Several studies have shown that systematic exercise intervention can effectively enhance muscle strength, joint flexibility, and reaction speed in older adults (Liu & Latham, 2009), especially some exercises specifically targeting balance function (e.g., tai chi, pilates, and balance training) have demonstrated good results in fall prevention (Voukelatos et al., 2007; Długosz-Boś et al., 2021). Although existing systematic reviews have confirmed the potential value of exercise interventions for fall prevention and balance function improvement in older adults, existing meta-analyses have mostly focused on the overall effect of exercise interventions, and lacked in-depth exploration of the heterogeneous effects of the relevant elements of the exercise program (single exercise time, exercise frequency, and exercise cycle), resulting in a lack of precise exercise program references in clinical practice. Accordingly, we conducted a systematic review and meta-analysis of randomized controlled trials (RCTs) evaluating exercise interventions for falls and balance in older adults, following evidence-based medicine guidelines. The aim was to quantitatively assess the overall effect of exercise intervention on fall efficacy and balance function in older adults; to clarify the differential effects of single exercise duration, frequency and cycle on the intervention effect through subgroup analysis; to propose optimization of exercise protocols, to make up for the shortcomings of the existing studies in terms of exercise dosage and individualized intervention protocols, and to explore more targeted and comprehensive exercise program, and to provide evidence and reference for researchers.

Survey Methodology

The review followed the PRISMA guidelines and the Cochrane Handbook for meta-analysis and systematic review (Higgins et al., 2019; Page et al., 2021). The research was registered on the International Prospective Register of Systematic Reviews (PROSPERO), identifier: CRD42024590937.

Inclusion and exclusion criteria

According to the PICOS principle, article inclusion criteria included:

(1) Population: older adults, aged ≥60, and without any disease.

(2) Intervention: exercise (tai chi chuan, baduanjin, etc.).

(3) Comparison: non-exercise interventions such as fall prevention lectures and blank controls.

(4) Outcomes: continuous variables: Modified Fall Efficacy Scale (MFES), Berg Balance Scale (BBS), Timed Up and Go Test (TUGT), the results presented as mean (M) ± standard deviation (SD). Dichotomous variable: number of falls (NF), extracted from the article.

(5) Type of study: randomized controlled trial (RCT).

Exclusion criteria: studies involving (1) older adults with neurological disorders (e.g., stroke, Parkinson’s, etc.), cardiovascular conditions (e.g., hypertension, etc.), or other diseases; (2) non-RCT, conference papers, abstracts, and review articles; (3) studies with outcomes that did not meet the specified criteria; (4) non-English or non-Chinese literature; (5) animal studies.

Literature search

Literature search as described above (Yu et al., 2024). Specifically, search in PubMed, Web of Science, Embase, Cochrane Library and CNKI databases. The retrieval strategy was based on MeSH subject words and free words with “AND” and “OR” linking, e.g.: (“older adults” OR “old people” OR “aged”) and (“exercise” OR “exercise intervention” OR “physical exercise” OR “sport” OR “physical activity” OR “exercise” OR “yoga”) and (“accidental falls” OR “fall” OR “balance” OR “dynamic balance” OR “postural balance”). The complete retrieval strategy is in the supplementary material. The retrieval period was from the database creation date to August 2024, and references to retrieved literature were backdated.

Literature screening

The literature was screened as described above (Yu et al., 2024). Specifically, the retrieved literature was imported into Endnote X9.1, after removing duplicates, two researchers (SNC and ML) independently screened the titles and abstracts according to the inclusion and exclusion criteria, and then read the full texts for further screening. If the results were consistent, the literature was included in this review, if not, it will be discussed with the 3rd researcher (HRY) until a consensus was reached. When discrepancies arose, the 3rd researcher (HRY) primarily consulted with SNC and ML based on established inclusion and exclusion criteria to determine whether studies in dispute should be included in this review. In practice, only a small number of discrepancies occurred during the screening process, all of which were resolved through discussion, and no studies were excluded solely due to unresolved disagreement.

Data extraction

Data were collected as previously described in Yu et al. (2024). Specifically, two researchers (SNC and ML) independently extracted data from the eligible literature using an agreed form. Disagreements were discussed with the 3rd researcher (JWZ) until consensus was reached. When discrepancies arise during data extraction, we resolve them by thoroughly reviewing the full text and conducting group discussions. If the issue remains unresolved, we contact the original author. The main components extracted were: (1) basic information about the included literature (first author, publication year, country); (2) subject characteristics (sample size, age, other characteristics); (3) intervention in the experimental group (exercise content, single exercise time, frequency, exercise cycle); (4) intervention in the control group; and (5) outcome indicators.

Quality assessment

Literature quality assessment as described above (Yu et al., 2024). Specifically, the methodological quality of the literature was evaluated using the Cochrane Risk Assessment Tool, which includes seven items: random sequence generation, Allocation concealment, blinding of participants and personnel, blinding of outcome assessment, incomplete outcome data, selective reporting, other bias. Each literature was assessed in three options: high risk, unclear risk, and low risk. It was independently by two researchers (SNC and ML), and when results were inconsistent, it was resolved by discussion with the 3rd researcher (HRY) until a consensus was reached.

Statistical analysis

Data analysis was described previously (Yu et al., 2024). Specifically, Review Manager 5.4 was used for methodological quality assessment of the literature, and Stata 17 was used for publication bias testing (including Egger’s test and Begg’s test), sensitivity analysis, pooled effect sizes, forest plotting, and subgroup analysis. The data used in this review are the change values of M and SD from baseline to endpoint. If it cannot be extracted directly, it is estimated according to the following formula: M = M2–M1 (M2 is the endpoint mean, M1 is the baseline mean); SD=SD12+SD22−2×Corr×SD1×SD2 (SD1 is the baseline SD, SD2 is the endpoint SD, Corr is the correlation coefficient between the baseline and endpoint scores, conservatively set at 0.5) (Follmann et al., 1992; Fukuta et al., 2016). Effect sizes were expressed using odds ratio (OR) for dichotomous variables and Hedges’ g (g) for continuous variables. Each effect size was provided with its 95% confidence interval (CI). I2 < 50% uses the fixed-effects model; I2 > 50% uses the random effects model to pooled effect sizes, then conduct subgroup analysis and sensitivity analysis (Higgins & Thompson, 2002). p < 0.05 was defined as statistically significant (Higgins et al., 2011).

Results

Review selection and characteristics

A total of 12,064 literatures were retrieved, 6,757 literatures remained after eliminating duplicates, 172 literatures were obtained by further screening based on title and abstract. Finally, the full text was read, 132 literatures were excluded due to non-compliance with outcome indicators, unavailability of full text, non-exercise intervention, non-RCT, and disease. Additionally, one study was obtained from the references cited in the included literature. A total of 38 literatures were finally included, including 22 English literatures and 16 Chinese literatures (Fig. 1).

Figure 1 Literature screening flowchart.

Results of the quality assessment of the included literature

The 38 included literature, all used randomized methods to allocate members of the intervention and control groups, provided complete data, reported results unselectively, and found no other bias. 22 literatures described procedures for allocating concealment, eight literatures used blinding for implementers, and all literatures were not blinded to the subjects (Fig. 2).

Figure 2 Literature quality evaluation.

Note. Birimoglu & Bilgili (2017); Byoung et al. (2012); Długosz-Boś et al. (2021); Donatoni, Shiel & McIntosh (2022); Du et al. (2020); Ehrari et al. (2020); Fatma & Bilgili (2023); Ferreira et al. (2022); Fuzhong et al. (2005); Gu et al. (2020); Hong et al. (2018); Hosseini et al. (2018); Huang et al. (2023); Hyuk & Lee (2020); Kovács et al. (2013); Lee & Lee (2017); Lee (2023); Liu et al. (2015); Liu (2020); Ma (2014); Ma & Zhang (2016); Ma et al. (2021); Ma et al. (2023); Patti et al. (2017); Pei et al. (2023); Pepera et al. (2021); Roller et al. (2018); Sadaqa et al. (2024); Shen, Lu & Fang (2022); Sitthiracha, Eungpinichpong & Chatchawan (2021); Song, Liu & Lv (2020); Sousa & Mendes (2015); Wu et al. (2017); Xing (2023); Yang et al. (2023); Yi & Yim (2021); Zhang et al. (2018); Zhou et al. (2020).

Characteristics of the included studies

There were 38 RCT. 10 literatures included MFES as an outcome indicator, six literatures included NF as an outcome indicator, 18 literatures included BBS as an outcome indicator, and 27 literatures included TUGT as an outcome indicator. The exercise content included tai chi chuan, baduanjin, etc.; the single exercise time ranged from 12–70 min; the frequency ranged from 1–5 times/week; the exercise cycle ranged from 4 weeks–1 year; the control group used fall prevention lectures and blank controls, etc. The basic characteristics of the included literatures are shown in Table 1.

Table 1 Characteristics of the included literature.

Note. Birimoglu & Bilgili (2017); Byoung et al. (2012); Długosz-Boś et al. (2021); Donatoni, Shiel & McIntosh (2022); Du et al. (2020); Ehrari et al. (2020); Fatma & Bilgili (2023); Ferreira et al. (2022); Fuzhong et al. (2005); Gu et al. (2020); Hong et al. (2018); Hosseini et al. (2018); Huang et al. (2023); Hyuk & Lee (2020); Kovács et al. (2013); Lee & Lee (2017); Lee (2023); Liu et al. (2015); Liu (2020); Ma (2014); Ma & Zhang (2016); Ma et al. (2021); Ma et al. (2023); Patti et al. (2017); Pei et al. (2023); Pepera et al. (2021); Roller et al. (2018); Sadaqa et al. (2024); Shen, Lu & Fang (2022); Sitthiracha, Eungpinichpong & Chatchawan (2021); Song, Liu & Lv (2020); Sousa & Mendes (2015); Wu et al. (2017); Xing (2023); Yang et al. (2023); Yi & Yim (2021); Zhang et al. (2018); Zhou et al. (2020).

First author	Country	Target sample	Total (intervention group/control group)	Intervention in the experimental group	Intervention in the control group	Frequency	Time	Exercise cycle	Outcome indicator	
Birimoğlu 2017	Turkish	Healthy older adults (age ≥ 65)	44 (20/24)	Tai Chi Chuan	Routine health guidance	2 times/week	30 min/time	12 weeks	MFES	
Byoung 2012	Korea	Healthy older adults (age ≥ 78)	78 (38/40)	Swiss Ball	Blank waiting list	2 times/week	30 min/time	12 weeks	TUGT, OLST	
Długosz 2021	Polish	Healthy older adults (age ≥ 60)	50 (30/20)	Pilates	Blank waiting list	2 times/week	45 min/time	3 months	TUGT, OLST	
Donatoni 2022	Irish	Healthy older adults (age ≥ 65)	61 (29/32)	Pilates	Blank waiting list	2 times/week	30 min/time	12 weeks	TUGT	
Du 2020	China	Healthy older adults (age 60–80)	157 (94/63)	Baduanjin	Health education	5 times/week	60 min/time	6 months	BBS, TUGT	
Ehrari 2020	Denmark	Healthy older adults (age ≥ 65)	26 (14/12)	Playful exercise	Blank waiting list	2 times/week	12 min/time	12 weeks	BBS	
Fatma 2023	Turkish	Healthy older adults (age ≥ 65)	56 (28/28)	Otago exercise	Blank waiting list	3 times/week	30 min/time	8 weeks	MFES, BBS	
Ferreira 2022	Brazilian	Healthy older adults (age ≥ 65)	49 (24/25)	Aquatic exercise	Health surveillance	2 times/week	60 min/time	16 weeks	TUGT	
Fuzhong 2005	USA	Healthy older adults (age 70–92)	256 (125/131)	Tai Chi Chuan	Blank waiting list	3 times/week	60 min/time	6 months	BBS, TUGT, NF	
Gu 2020	China	Older adults who fear falling	60 (30/30)	Otago exercise	Health education	3 times/week	30 min/time	24 weeks	MFES, BBS, TUGT	
Hong 2018	China	Healthy older adults (age ≥ 65)	72 (37/35)	Tai Chi Chuan	Health education	3 times/week	45 min/time	6 months	MFES, NF	
Hosseini 2018	Iranian	Healthy older adults (age 60–80)	60 (30/30)	Tai Chi Chuan	Blank waiting list	2 times/week	60 min/time	8 weeks	TUGT	
Huang 2023	China	Healthy older adults (age>65)	35 (17/18)	Baduanjin	Health education	3 times/week	30 min/time	12 weeks	TUGT, MFES	
Hyuk 2020	Korea	Healthy older adults (age ≥ 75)	30 (15/15)	Body vibration exercise	Blank waiting list	3 times/week	25 min/time	4 weeks	BBS, TUGT	
Kovacs 2013	Hungary	Healthy older adults (age ≥ 60)	76(38/38)	Physical exercise	Blank waiting list	2 times/week	60 min/time	25 weeks	TUGT	
Lee 2017	Korea	Older adults with experience of falls (age ≥ 65)	54 (27/27)	Balance Training	Fall prevention talk	1 times/week	60 min/time	4 weeks	BBS, TUGT, OLST	
Lee 2023	Korea	Healthy older adults (age ≥ 75)	57 (28/29)	Sports Game	Blank waiting list	3 times/week	50 min/time	8 weeks	MFES, BBS, TUGT, OLST	
Liu 2015	China	Healthy older adults (age ≥ 60)	95(47/48)	Baduanjin	Health education		30 min/time	12 weeks	MFES	
Liu 2020	China	Healthy older adults (age 65–96)	64 (32/32)	Resistance training	Health education	3 times/week	30 min/time	8 weeks	BBS	
Ma 2014	China	Healthy older adults (age 60–70)	72 (36/36)	Tai Chi Ball	Blank waiting list	3 times/week	60 min/time	12 months	TUGT	
Ma 2016	China	Healthy older adults (age 60–65)	42 (22/20)	Baduanjin	Blank waiting list	3 times/week	60 min/time	6 months	TUGT, OLST	
Ma 2021	China	Healthy older adults (age ≥ 60)	50 (25/25)	Physical exercise	Blank waiting list			12 months	TUGT, OLST, NF	
Ma 2023	China	Healthy older adults (age ≥ 60)	78 (38/40)	Otago exercise	Routine falls prevention care			12 weeks	TUGT	
Patti 2017	Italy	Healthy older adults (age 65–85)	92 (49/43)	Physical exercise	Blank waiting list	2 times/week	70 min/time	13 weeks	BBS	
Pei 2023	China	Healthy older adults (age ≥ 60)	79(39/40)	Otago exercise	Blank waiting list	3 times/week	45 min/time	12 weeks	MFES, BBS, TUGT	
Pepera 2021	Greece	Healthy older adults (age ≥ 65)	40 (20/20)	multicomponent exercise program	Blank waiting list	2 times/week	45 min/time	8 weeks	BBS, TUGT	
Roller 2018	USA	Older adults with experience of falls (age ≥ 65)	55 (27/28)	Pilates	Blank waiting list	1 times/week	45 min/time	10 weeks	BBS, TUGT	
Sadaqa 2024	Hungary	Healthy older adults (age ≥ 65)	24 (12/12)	Aerobic exercise	Blank waiting list	2 times/week	60 min/time	12 weeks	TUGT, OLST	
Shen 2022	China	Healthy older adults (age ≥ 60)	430 (215/215)	Tai Chi Chuan	Health education	4 times/week	30 min/time	8 weeks	MFES, NF	
Sitthiracha 2021	Thailand	Healthy older adults (age ≥ 65)	60 (30/30)	Aerobic exercise	Blank waiting list	5 times/week	45 min/time	8 weeks	TUGT, OLST	
Song 2020	China	Healthy older adults (age 62–85)	110 (55/55)	Balance training	Routine health guidance			12 months	BBS, NF	
Sousa 2015	Portugal	Healthy older adults (age 65–80)	22 (12/10)	Resistance training	Blank waiting list	2 times/week	20 min/time	12 weeks	TUGT	
Wu 2017	China	Older adults with experience of falls (age 65–80)	120 (60/60)	Baduanjin	Blank waiting list		30 min/time	3 months	BBS, TUGT	
Xing 2023	China	Healthy older adults (age ≥ 60)	100 (50/50)	Towel exercise	Fall prevention talk	5 times/week	30 min/time	12 months	NF	
Yang 2023	USA	Healthy older adults (age ≥ 65)	42 (22/20)	Body vibration exercise	Blank waiting list	3 times/week	45 min/time	8 weeks	BBS	
Yi 2021	Korea	Healthy older adults (age ≥ 65)	70 (35/35)	Physical exercise	Blank waiting list	2 times/week	40 min/time	8 weeks	TUGT	
Zhang 2018	China	Healthy older adults (age 65–75)	120 (60/60)	Obstacle-crossing training	Fall prevention talk			6 months	TUGT, MFES	
Zhou 2020	China	Healthy older adults (age ≥ 80)	30 (15/15)	Aerobic exercise	Health education	5 times/week	30 min/time	3 months	TUGT	

Meta-analysis results

Falls indicators

In this review, 10 literatures assessed MFES scores, with a total sample size of 1,048 subjects (521 in the intervention group and 527 in the control group). The overall heterogeneity test (I2 = 86.11%, p = 0.00) indicated that there was heterogeneity among multiple studies, so using the random effects model pooled the effect sizes: (g = 1.01, 95% CI [0.63–1.40], p = 0.00), which was statistically significant (Fig. 3), showed that exercise intervention can improve fall efficacy in older adults.

There are six literatures assessed the effect of exercise intervention on NF in older adults, with a total sample size of 950 subjects (477 in the intervention group and 473 in the control group). The overall heterogeneity test (I2 = 0.00%, p = 0.83) showed no heterogeneity among multiple studies, so using fixed effects models for meta-analysis: (OR = 0.32, 95% CI [0.20–0.51], P = 0.00) was statistically significant (Fig. 4), showed that the intervention group was 32% as likely to fall as the control group, and that the intervention group was significantly less likely to have a fall than the control group.

Figure 3 Effect of exercise intervention on MFES.

Note. Birimoglu & Bilgili, 2017; Fatma & Bilgili, 2023; Gu et al., 2020; Hong et al., 2018; Huang et al., 2023; Lee, 2023; Liu et al., 2015; Pei et al., 2023; Shen, Lu & Fang, 2022; Zhang et al., 2018.

Figure 4 Effect of exercise intervention on the number of falls.

Note. Fuzhong et al., 2005; Hong et al., 2018; Ma et al., 2021; Shen, Lu & Fang, 2022; Song, Liu & Lv, 2020; Xing, 2023.

Balanced function indicators

In this review, 16 literatures assessed BBS scores, with a total sample size of 1,298 subjects (665 in the intervention group and 633 in the control group). The overall heterogeneity test (I2 = 80.71%, p = 0.00) indicated that there was heterogeneity among multiple studies, so using the random effects model pooled the effect sizes: (g = 0.89, 95% CI [0.61–1.17], p = 0.00), which was statistically significant (Fig. 5), showed that exercise intervention can improve the balance ability in older adults.

Figure 5 Effect of exercise intervention on BBS.

Note. Du et al., 2020; Ehrari et al., 2020; Fatma & Bilgili, 2023; Fuzhong et al., 2005; Gu et al., 2020; Hyuk & Lee, 2020; Lee & Lee, 2017; Lee, 2023; Liu, 2020; Patti et al., 2017; Pei et al., 2023; Pepera et al., 2021; Roller et al., 2018; Song, Liu & Lv, 2020; Wu et al., 2017; Yang et al., 2023.

There are 27 literatures assessed TUGT outcomes, with a total sample size of 1,885 subjects (956 in the intervention group and 929 in the control group).The overall heterogeneity test (I2 = 68.77%, p = 0.00) indicated that there was heterogeneity among multiple studies, so using the random effects model pooled the effect sizes: (g = −0.62, 95% CI [−0.79,−0.44], p = 0.00) which was statistically significant (Fig. 6), showed that exercise intervention can improve the TUGT scores in older adults.

Figure 6 Effect of exercise intervention on TUGT.

Note. Byoung et al., 2012; Długosz-Boś et al., 2021; Donatoni, Shiel & McIntosh, 2022; Du et al., 2020; Ferreira et al., 2022; Fuzhong et al., 2005; Gu et al., 2020; Hosseini et al., 2018; Huang et al., 2023; Hyuk & Lee, 2020; Kovács et al., 2013; Lee & Kim, 2017; Lee, 2023; Ma, 2014; Ma & Zhang, 2016; Ma et al., 2021; Ma et al., 2023; Pei et al., 2023; Pepera et al., 2021; Roller et al., 2018; Sadaqa et al., 2024; Sitthiracha, Eungpinichpong & Chatchawan, 2021; Sousa & Mendes, 2015; Wu et al., 2017; Yi & Yim, 2021; Zhang et al., 2018; Zhou et al., 2020.

Subgroup analysis of moderators

This review conducted a subgroup analysis, and the results are shown in Table 2.

Table 2 Subgroup analysis of MFES, BBS, TUGT in older adults undergoing exercise intervention.

Moderators	Outcome	Homogeneity test	Category	Number of literatures	Number of samples	Effect size and 95% CI	Two-tailed test	
		Q	P	I2 (%)					Q	P	
Single exercise time	MFES	55.19	0.000	86.11	≤30 min/time	6	720	0.87 (0.57, 1.17)	11.48	0.000	
>30 min/time	4	325	1.01 (0.63, 1.40)	37.94	0.008	
BBS	44.05	0.000	71.05	≤30 min/time	7	435	1.14 (0.80, 1.48)	14.91	0.000	
>30 min/time	8	753	0.58 (0.36, 0.79)	11.68	0.000	
TUGT	58.47	0.000	60.99	≤30 min/time	9	481	−0.98 (−1.34, −0.62)	26.43	0.000	
>30 min/time	15	1,122	−0.45 (−0.63, −0.27)	28.36	0.000	
Exercise frequency	MFES	46.32	0.000	84.92	<3 times/week	1	44	1.61 (0.92, 2.30)	−0.00	0.000	
3 times/week	6	441	1.05 (0.47, 1.63)	41.15	0.000	
>3 times/week	2	465	0.90 (0.71, 1.09)	0.49	0.000	
BBS	39.26	0.000	70.26	<3 times/week	5	267	0.38 (0.14,0.62)	1.38	0.002	
3 times/week	8	644	0.97 (0.63, 1.32)	24.65	0.000	
>3 times/week	1	157	1.04 (0.70, 1.38)	−0.00	0.000	
TUGT	58.46	0.000	62.30	<3 times/week	12	639	−0.59 (−0.83, −0.34)	24.90	0.000	
3 times/week	7	596	−0.69 (−1.03, −0.35)	16.81	0.000	
>3 times/week	4	282	−0.60 (−1.15, −0.05)	14.12	0.033	
Study duration	MFES	55.19	0.000	86.11	<12 weeks	3	540	0.70 (0.37, 1.03)	3.94	0.000	
12–23 weeks	4	253	1.25 (0.53, 1.96)	24.46	0.001	
≥24 weeks	3	252	1.11 (0.27, 1.96)	20.73	0.010	
BBS	71.27	0.000	80.71	<12 weeks	9	549	0.73 (0.42, 1.04)	23.54	0.000	
12–23 weeks	3	197	0.82 (0.03, 1.61)	12.84	0.041	
≥24 weeks	4	583	1.25 (0.64, 1.85)	30.40	0.000	
TUGT	82.46	0.000	68.77	<12 weeks	9	546	−0.54 (−0.76, −0.33)	12.27	0.000	
12–23 weeks	10	506	−0.55 (−0.96, −0.15)	46.24	0.008	
≥24 weeks	8	833	−0.83 (−1.19, −0.47)	29.46	0.000	

Single exercise time

Of the 10 literatures assessed MFES, four literatures single exercise time of >30 min showed a pooled effect size g = 1.01, 95% CI [0.63–1.40], p = 0.008. Six literatures single exercise time of ≤30 min showed a pooled effect size g = 0.87, 95% CI [0.57–1.17], p < 0.001. Therefore, it is inferred that exercise intervention>30 min per session has a larger impact on fall efficacy in older adults.

Of the 16 literatures assessed BBS, 1literature didn’t specify the single exercise time and was thus excluded from the subgroup analysis. Eight literatures single exercise time of >30 min showed a pooled effect size g = 0.58, 95% CI [0.36–0.79], p < 0.001. Seven literatures single exercise time of ≤ 30 min showed a pooled effect size g = 1.14, 95% CI [0.80–1.48], p < 0.001. Therefore, it is inferred that exercise intervention ≤30 min per session has a larger impact on balance ability in older adults.

Of the 27 literatures assessed TUGT, three literatures didn’t specify the single exercise time and was thus excluded from the subgroup analysis. 15 literatures single exercise time of >30 min showed a pooled effect size g = −0.45, 95% CI [−0.63, −0.27), p < 0.001. Nine literatures single exercise time of ≤30 min showed a pooled effect size g = −0.98, 95% CI [−1.34, −0.62], p < 0.001. Therefore, it is inferred that exercise intervention ≤30 min per session have the greatest impact on TUGT scores in older adults.

Exercise frequency

Of the 10 literatures assessed MFES, one literature didn’t specify the exercise frequency and was thus excluded from the subgroup analysis. One literature exercise frequency of <3 times/week showed the effect size g = 1.61, 95% CI [0.92–2.30], p < 0.001. Six literatures exercise frequency of 3times/week showed a pooled effect size g = 1.05, 95% CI [0.47–1.63], p < 0.001. Two literatures exercise frequency of >3times/week showed a pooled effect size g = 0.90, 95% CI [0.71–1.09), p < 0.001. Because there was only one literature exercise frequency of <3 times/week, it was not comparable. Therefore, it is inferred that exercise frequency of 3 times/week has a larger impact on fall efficacy in older adults.

Of the 16 literatures assessed BBS, two literatures didn’t specify the exercise frequency and was thus excluded from the subgroup analysis. Five literatures exercise frequency of <3 times/week showed a pooled effect size g = 0.38, 95% CI [0.14–0.62], p = 0.002. Eight literatures exercise frequency of 3 times/week showed a pooled effect size g = 0.97, 95% CI [0.63–1.32], p < 0.001. One literature exercise frequency of >3 times/week showed the effect size g = 1.04, 95% CI [0.70–1.38], p < 0.001. Because there was only one literature exercise frequency of >3 times/week, it was not comparable. Therefore, it is inferred that exercise frequency of 3 times/week has a larger impact on balance ability in older adults.

Of the 27 literatures assessed TUGT, four literatures didn’t specify the exercise frequency and was thus excluded from the subgroup analysis. 12 literatures exercise frequency of <3 times/week showed a pooled effect size g = −0.59, 95% CI [−0.83, −0.34], p < 0.001. Seven literatures exercise frequency of 3 times/week showed a pooled effect size g = −0.69, 95% CI [−1.03, −0.35], p < 0.001. Four literatures exercise frequency of >3 times/week showed a pooled effect size g =−0.60, 95% CI [−1.15, −0.05], p = 0. 033.Therefore, it is inferred that exercise frequency of 3 times/week has a larger impact on TUGT scores in older adults.

Study duration

Of the 10 literatures assessed MFES, three literatures exercise cycle of <12 weeks showed a pooled effect size g = 0.70, 95% CI [0.37–1.03], p < 0.001. Four literatures exercise cycle of 12–23 weeks showed a pooled effect size g = 1.25, 95% CI [0.53–1.96], p = 0.001. Three literatures exercise cycle of ≥24 weeks showed a pooled effect size g = 1.11, 95% CI [0.27–1.96], p = 0.010. Therefore, it is inferred that exercise cycle of 12–23 weeks has a larger impact on fall efficacy in older adults.

Of the 16 literatures assessed BBS, nine literatures exercise cycle of <12 weeks showed a pooled effect size g = 0.73, 95% CI [0.42–1.04], p < 0.001. Three literatures exercise cycle of 12-23 weeks showed a pooled effect size g = 0.82, 95% CI [0.03–1.61], p = 0.041. Four literatures exercise cycle of ≥24 weeks showed a pooled effect size g = 1.25, 95% CI [0.64–1.85], p < 0.001. Therefore, it is inferred that exercise cycle of ≥24 weeks has a larger impact on balance ability in older adults.

Of the 27 literatures assessed TUGT, nine literatures exercise cycle of <12 weeks showed a pooled effect size g =−0.54, 95% CI [−0.76, −0.33], p < 0.001. 10 literatures exercise cycle of 12–23 weeks showed a pooled effect size g =−0.55, 95% CI [−0.96, −0.15], P = 0.008. 8 literatures exercise cycle of ≥24 weeks showed a pooled effect size g =−0.83, 95% CI [−1.19, −0.47], p<0. 001. Therefore, it is inferred that exercise cycle of ≥24 weeks has a larger impact on TUGT scores in older adults.

Publication bias test and sensitivity analysis

In this review, funnel plots were drawn for the indices MFES, BBS, and TUGT to analyze their symmetry and assess the presence of publication bias (Figs. 7, 8 and 9). The distribution of data points in the three plots is asymmetric, suggesting the presence of some degree of publication bias in the included literature.

Figure 7 Funnel plot (MFES).

Figure 8 Funnel plot (BBS).

Figure 9 Funnel plot (TUGT).

We conducted sensitivity analyses separately for the literature included in MFES, BBS, and TUGT indicators (Figs. 10, 11 and 12). The literature incorporated in each of these three indicators exhibited good stability; exclusion of any individual study from any one indicator did not produce significant changes in the results for that respective indicator, confirming that the meta-analysis findings on the effects of exercise interventions on falls and balance function in older adults are reliable.

Figure 10 Sensitivity analysis (MFES).

Note. Birimoglu & Bilgili, 2017; Fatma & Bilgili, 2023; Gu et al., 2020; Hong et al., 2018; Huang et al., 2023; Lee, 2023; Liu et al., 2015; Pei et al., 2023; Shen, Lu & Fang, 2022; Zhang et al., 2018.

Figure 11 Sensitivity analysis (BBS).

Note. Du et al., 2020; Ehrari et al., 2020; Fatma & Bilgili, 2023; Fuzhong et al., 2005; Gu et al., 2020; Hyuk & Lee, 2020; Lee & Lee, 2017; Lee, 2023; Liu, 2020; Patti et al., 2017; Pei et al., 2023; Pepera et al., 2021; Roller et al., 2018; Song, Liu & Lv, 2020; Wu et al., 2017; Yang et al., 2023.

Figure 12 Sensitivity analysis (TUGT).

Note. Byoung et al., 2012; Długosz-Boś et al., 2021; Donatoni, Shiel & McIntosh, 2022; Du et al., 2020; Ferreira et al., 2022; Fuzhong et al., 2005; Gu et al., 2020; Hosseini et al., 2018; Huang et al., 2023; Hyuk & Lee, 2020; Kovács et al., 2013; Lee & Lee, 2017; Lee, 2023; Ma, 2014; Ma & Zhang, 2016; Ma et al., 2021; Ma et al., 2023; Pei et al., 2023; Pepera et al., 2021; Roller et al., 2018; Sadaqa et al., 2024; Sitthiracha, Eungpinichpong & Chatchawan, 2021; Sousa & Mendes, 2015; Wu et al., 2017; Yi & Yim, 2021; Zhang et al., 2018; Zhou et al., 2020.

Quality assessment of the included article

Statistical methods and outcome indicators were identified in all 38 articles included. However, only nine articles used the blinding method for implementors, and all articles didn’t use the blinding method for subjects. This may affect the quality of articles but didn’t affect the effectiveness of exercise intervention on falls and balance function in older adults. This would have facilitated a clear understanding of the purpose of the experiment by the subjects and facilitated the smooth implementation of the experiment. Therefore, not using blinding for all subjects and some of the experiment’s implementers didn’t affect the overall experimental results.

The reasons for the asymmetry in the funnel plot of the three indicators may be that (1) studies with nonsignificant results may be less likely to be published, which may lead to overestimation of the intervention effect; (2) there is a large variability in the sample sizes of the included studies, and smaller sample sizes are more prone to randomization error, which may lead to publication bias; (3) the studies included in this analysis vary greatly in terms of intervention duration, ranging from a minimum of 4 weeks to a maximum of 1 year, baseline information across studies could differ significantly, potentially introducing bias.

Discussion

Analysis of overall effect

Meta-analysis of this review showed that exercise intervention had a significant improvement on falls efficacy (MFES: (g =1.01, 95% CI [0.63–1.40], p = 0.00)) and number of falls (OR=0.32, 95% CI [0.20–0.51], P = 0.00) in older adults. Also, balance function (BBS: (g =0.89, 95% CI [0.61–1.17], p = 0.00); TUGT: (g =−0.62, 95% CI [−0.79, −0.44], p = 0.00)) of the older adults was significantly improved after the exercise intervention. This finding is consistent with the findings of previous meta-analyses, but some differences remain, specifically: the improvement in MFES in this review was higher than that of Huang et al. (2020). (g = 0.88), but slightly lower than that of Guo et al. (2021) (g = 1.12). This difference may be related to the different types of exercise included in the studies, for example, Huang et al. only included studies focusing on tai chi, whereas the present review covered a variety of exercise forms such as tai chi, aerobic exercise, and balance training. In addition, the magnitude of improvement in TUGT test scores was like Huang et al. (2020). (g = −0.71). Therefore, this review further validates the effectiveness of exercise intervention in enhancing fall efficacy and balance function in older adults.

Aging is associated with gradual declines in skeletal muscle strength, cardiovascular function, vision, vestibular function, and proprioception. These changes reduce coordination and slow postural responses. They also contribute to cognitive decline; together, these effects impair balance and increase fall risk (Segev-Jacubovski et al., 2011). Exercise interventions can significantly enhance muscle strength (Pepera et al., 2023), particularly in the hip, knee, and ankle regions. This increased strength improves balance and reduces fall risk, especially in unstable environments or unexpected situations (Sherrington et al., 2011). Aerobic exercise enhances cardiorespiratory function and improves blood circulation in older adults, which positively impacts their responsiveness (Li, Li & Li, 2021). Cardiovascular health improvements help mitigate symptoms such as fainting and dizziness that often occur post-fall, further reducing the adverse outcomes associated with falls (Zhou & Yu, 2006). Additionally, frequent postural adjustments during exercise (e.g., dynamic balance training and gait exercises) enhance proprioception and vestibular function. These activities promote adaptive changes in the nervous system, enabling older adults to better perceive shifts in body posture and environmental changes, facilitating quicker responses to maintain balance and reduce the incidence of falls (Granacher, Gruber & Gollhofer, 2009). Complex training modalities have been shown to enhance neural plasticity and to improve cognitive functions including attention, spatial awareness, and decision-making in older adults. These improvements enable them to better detect potential hazards and make necessary adjustments to prevent falls (Liu & Latham, 2009). Sustained participation in physical activity builds mobility confidence and reduces fear of falling. Increased confidence promotes engagement in daily activities, which further improves balance and motor skills and creates a positive feedback loop for physical and psychological well-being (Skelton & Dinan, 1999).

Single exercise time

Subgroup analysis of the MFES indicators on single exercise time showed that the intervention effect was more effective for a single exercise time >30 min. The subgroup analysis of the BBS and TUGT on single exercise time was consistent with the result that the intervention effect was more effective for ≤30 min. Longer exercise time provides older adults with sufficient opportunities for comprehensive strength training and complex movement practice, which not only improves physical function and muscle strength, but also enhances their confidence and sense of control over their abilities, contributing to improved fall efficacy (Hamed et al., 2018; Mansson et al., 2020). Both BBS and TUGT are measures of balance function, and short, efficient exercise intervention may be more targeted to balance training, such as static and dynamic balance exercises (Granacher, Gruber & Gollhofer, 2009). This duration of exercise reduces the risk of physical overexertion in older adults while ensuring that they can focus on balance control skills, contributing to core stability, postural alignment, and gait control, which can rapidly improve balance in older adults (Shumway-Cook et al., 1997).

Exercise frequency

The results of the subgroup analyses on exercise frequency were consistent across the three indicators, the better effect of the intervention was 3 times/week, i.e., 3 times/week exercise workouts were more likely to achieve the best results in improving fall efficacy and balance function in older adults. A moderate frequency of exercise can provide older adults with adequate physical stimulation while giving the body sufficient recovery time (Suen et al., 2024). It not only promotes the improvement of muscle strength and neuromuscular coordination, but also avoids physical fatigue and exercise burnout due to excessive exercise, which is conducive to the steady improvement of older adult’s mobility, enhancing their confidence in independent activities and improving their sense of fall efficacy (Sherrington et al., 2020). Exercise intervention for older adults’ balance function should not be too frequent, and too high an exercise frequency is not conducive to focusing on balance training, e.g., although the TUGT is closely related to balance function, it also requires older adults to have better reaction speed and lower limb strength, and relying solely on higher frequency training may not be sufficient to significantly improve the results of this test (Sherrington et al., 2008). Appropriate exercise frequency can gradually enhance the agility of neuromuscular response and proprioception, strengthen postural adjustment and gait stabilization, ensure that older adults can maintain physical stability and flexibility in different environments, and improve older adult’s balance function while satisfying the effect of balance function training (Buchner et al., 1997).

Exercise cycle

Subgroup analyses of the MFES indicator on exercise cycle showed that exercise intervention lasting 12–23 weeks was more effective. Subgroup analyses of the BBS and TUGT on exercise cycle were consistent with the results that exercise intervention lasting ≥24 weeks was more effective. Much of the improvement in fall efficacy in older adults comes from increased confidence in voluntary activity; therefore, shorter cycles of exercise intervention allow older adults to perceive changes in their bodies in a shorter period, especially increased skeletal muscle strength (Gillespie et al., 2012). This positive feedback improves older adults’ ability to move autonomously, increase their confidence in doing so, positively affect their daily lives, reduce fear of falling, and contribute to fall efficacy (Lee & Kim, 2017). However, the improvement of balance in older adults may require long cycles of exercise intervention, and longer cycles of exercise intervention provide older adults with sufficient time to consolidate neuromuscular adaptive changes, which contributes to the stable development of balance function (Bates et al., 2018). Long-term exercise intervention help older adults establish a more robust postural control system, which not only facilitates the improvement of static balance, but also provides better support and protection for older adults in dynamic activities (Ercan Yildiz et al., 2024). At the same time, long-term exercise intervention is conducive to the development of good exercise habits in older adults, which promotes the long-term steady improvement of balance function (Okubo, Schoene & Lord, 2017).

Strengths of this review

First, this review focused on introducing refined subgroup analyses to dissect the effects of single exercise time, exercise frequency, and exercise cycle on fall efficacy and balance function in older adults, which have not been extensively explored in previous meta-analyses, and which may provide more specific for the implementation or development of exercise intervention programs. Second, although this meta-analysis is consistent with the results of most of the existing articles that validate the effectiveness of exercise interventions on falls and balance function in older adults. However, compared with other research, this review excluded combined interventions, only analyzed exercise interventions, and included all exercise types, which is more helpful in evaluating the final effect of exercise, and the results have a higher reliability. Thirdly, the inclusion of newer studies and the absence of obvious signs of bias in the funnel plots of all included studies, plus the high quality of the evidence and extensive data analysis, is another strength of this review.

Limitations and perspectives

This meta-analysis has certain limitations that we aim to address in future research and practice. First, due to the specificity of the older adult population and the nature of the exercise intervention, blinding was not applied to some implementers and all participants, introducing a risk of bias that may have affected the quality assessment. Future studies should adhere strictly to randomized controlled trial guidelines to ensure result reliability. Second, none of the included studies involved participants with diseases or cognitive impairments, which may limit the generalizability of the findings to populations with such conditions. Future research should focus on developing tailored exercise intervention for older adults with varying health conditions, including diseases and cognitive impairments, to achieve more comprehensive outcomes. Third, although the present review indicates more optimal exercise time, frequency, and duration. These are the key elements to achieve precise exercise interventions, but the current empirical research on the optimal ‘dose–response’ relationship remains insufficient. Therefore, there is an urgent need for more methodologically rigorous, high-quality studies, including large-sample randomized controlled trials and adequate follow-up surveys, to provide a reliable basis for the formulation of more scientific and standardized exercise intervention protocols. Lastly, while the reviewed studies utilized different exercise types, each study examined only a single intervention. Future investigations should explore the synergistic effects of combining exercise with other strategies, such as cognitive training, to develop integrated intervention and continuously optimize their effectiveness.

Conclusions and Recommendations

Exercise interventions are effective in improving fall efficacy, reducing the probability of falling, improving balance function, and playing a significant role in preventing falls in older adults. Key factors influencing the success of these interventions include the duration of individual exercise sessions, frequency of exercise, and the overall intervention duration. The results of this meta-analysis suggest that for improving fall efficacy, exercise sessions lasting >30 min, performed 3 times per week for 12–23 weeks, are more likely to yield optimal results. For improving balance function, exercise sessions lasting >30 min, performed 3 times per week for at least 24 weeks, are more effective.

Healthcare professionals and fitness trainers should develop precise exercise programs for older adults based on their specific physical conditions. When developing exercise programs, factors such as single exercise time, frequency, and cycle should be considered. During the exercise process, it is necessary to ensure the accuracy of the movements of the elderly, strengthen supervision and make timely adjustments to ensure the safety of the exercise.

Supplemental Information

Supplemental Information 1 Retrieval strategy

Supplemental Information 2 PRISMA 2020 Checklist

We thank everyone who contributed to this study.

Additional Information and Declarations

Competing Interests

Author Contributions

Data Availability

The authors declare there are no competing interests.

Haoran Yu conceived and designed the experiments, performed the experiments, analyzed the data, prepared figures and/or tables, authored or reviewed drafts of the article, and approved the final draft.

Jianwei Zhong conceived and designed the experiments, performed the experiments, analyzed the data, prepared figures and/or tables, authored or reviewed drafts of the article, and approved the final draft.

Min Li conceived and designed the experiments, performed the experiments, prepared figures and/or tables, authored or reviewed drafts of the article, and approved the final draft.

Shuainan Chen conceived and designed the experiments, performed the experiments, analyzed the data, authored or reviewed drafts of the article, and approved the final draft.

The following information was supplied regarding data availability:

This is a systematic review/meta-analysis.

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
