# Peer review of "Effects of exercise intervention on falls and balance function in older adults: a systematic review and meta-analysis"

_PeerJ, doi:10.7717/peerj.20190_

## Round 0.1 · original submission · Major Revisions

Thank you for your submission. The reviewers have identified a number of concerns that must be addressed. Please also check formatting and spacing.

Reviewer 1 ·

Basic reporting

In the abstract, please use the full term of the tests described before moving to abbreviations. For example: MFES, the BBS score.
Consider adding more PICO information in your abstract. For example: Objective: To systematically review the effects of an exercise intervention on falls and balance function in older adults (aged >60 years) without diagnosed diseases."

The introduction is well-written. There are some typos in the articles relating to formatting eg line 160; '5times/week'; the exercise cycle ranged from 4weeks 'to1' year - needs spacing. There is also inconsistent line spacing thought-out that needs reviewing.

In general replace the word "literature" with articles; For example on line 144:"A total of 12064 articles were retrieved, 6757 articles remained after eliminating 145 duplicates, 172 articles were obtained by further screening based on title and abstract.

I would move the first section of the discussion in line 258 Quality assessment of the included literature to the end of the result section.

I think the discussion needs a re-write to increase clarity of the study results:
Suggested Approach for Rewriting the Discussion:
1. Start with a Clear Summary of Key Findings:
Provide a concise overview of the study's primary results, including statistical outcomes and their implications.
Clearly link these findings to the research question or hypothesis.

2.Compare Results with Existing Literature:
Highlight how the findings align with or differ from previous studies, including meta-analyses.
Use direct comparisons (e.g., "This study's findings are consistent with X, which reported a similar reduction in falls due to exercise interventions").

Continue with Insights based on the 'Time, frequency and (cycle) exercise duration' section and conclude with 'Limitations and perspective' section:

Experimental design

The method section could be written using the PICO format with subheadings and a description of the methodology. Currently, the methods section does not describe the outcome used in the systematic review.

In line 232 please us the term Study duration instead of exercise cycle. The term "exercise cycle" might imply a specific type of exercise (e.g., cycling) or a phase within a larger program (e.g., a single cycle of periodized training). Hence, it’s less appropriate for referring to the overall length of a study or intervention.

I would also consider re-writing the conclusion based on the following to improve th readability of the article:
Exercise interventions effectively improve fall efficacy and balance function, reduce the number of falls, and play a significant role in preventing falls in older adults. Key factors influencing the success of these interventions include the duration of individual exercise sessions, frequency of exercise, and the overall intervention duration.

The results of this meta-analysis suggest that for improving fall efficacy, exercise sessions lasting >30 minutes, performed 3 times per week for 12–23 weeks, are more likely to yield optimal results. For improving balance function, exercise sessions lasting >30 minutes, performed 3 times per week for at least 24 weeks, are more effective.

Validity of the findings

In the Abstract and main article results section, there appears to be an inconsistency between p-value and CI: If the p-value is 0.00 (indicating significance) but the CI includes 1, there might be a reporting or methodological inconsistency. Typically, a statistically significant result would have a CI that does not include 1. Additionally, there needs to be a written interpretation of the statistics. For example: the number of falls [OR=1.18,95%CI(0.71,1.65),P=0.00] An OR of 1.18 indicates a 18% increase in the odds of falls associated with the variable compared to the reference group. However, the authors have reported it that showed that exercise intervention can effectively reduce the number of falls in older adults.

Be careful not to overstate findings. For example, in the subgroup analysis comparing the impact on MFES between exercise sessions lasting >30 minutes and <30 minutes, it is more accurate to write that an exercise intervention lasting >30 minutes per session has a larger (not greatest) impact on fall efficacy in older adults.

Reviewer 2 ·

Basic reporting

This is a really important and well-written paper. However there are a few amendments.

Experimental design

Abstract:
Please write what MFES score is.

Introduction:
- What is the main aim of this study. There are a lot of systematic on this field. What gap this paper comes to cover.


Results
"172 literatures were obtained by further screening based on title and abstract" Say more explain it.

'A total of
148 37 literatures were finally included, including 21 English literatures and 16 Chinese literatures'
Should be more.
There are papers that havent been included ie.
-Controlled Trial of Group Exercise Intervention for Fall Risk Factors Reduction in Nursing Home Residents.doi:10.1017/S0714980822000265
-Effects of multicomponent exercise training intervention on hemodynamic and physical function in older residents of long-term care facilities: A multicenter randomized clinical controlled trial. https://doi.org/10.1016/j.jbmt.2021.07.009

Line 211:
p=0.000, better write p<0.001

Validity of the findings

Discussion
Strength of study?
Future research?

---

## Round 0.2 · Minor Revisions

The manuscript requires revisions to enhance its clarity, consistency, and overall quality.

Specific Points for Revision
• Conflict and Disagreement Reporting: Please clarify whether any conflicts arose during the screening process. Additionally, the manuscript should detail any disagreements that occurred during data extraction. Providing this information will increase the transparency and rigor of your methodology.

• Terminology Consistency: For clarity and to avoid confusion, please use the term "study" exclusively when referring to the individual research articles you included in your analysis. The word "review" should be used when referencing your manuscript or the systematic process you conducted. This distinction is critical for the reader to understand which body of work you are discussing at any given point. For example, in line 224, please change "study" to "review."

• Reference and Citation: The citation for Yu et al. (2024) appears in several places to support different methodological steps (e.g., lines 130, 140, 148, 158). Please provide a clear justification for why this single reference is cited for multiple stages of your methodology, including literature search, screening, data extraction, and quality assessment.

• Grammar and Formatting: The manuscript would benefit significantly from comprehensive proofreading and editing, preferably by a native English speaker. There are several typographical and grammatical errors, such as:
o Spacing: Ensure correct spacing around brackets (e.g., a space before the opening bracket) and statistical symbols (e.g., spaces both before and after).
o Punctuation: Confirm that a space follows all punctuation marks.
o Sentence Structure: Correct incomplete sentences. For example, line 188 (Obtain 1 article from another.) should be revised for clarity and grammatical completeness.

**Language Note:** The Academic Editor has identified that the English language must be improved. PeerJ can provide language editing services - please contact us at [email protected] for pricing (be sure to provide your manuscript number and title). Alternatively, you should make your own arrangements to improve the language quality and provide details in your response letter. – PeerJ Staff

Reviewer 2 ·

Basic reporting

Most comments were addressed

Experimental design

Well defined

Validity of the findings

No comment

---

## Round 0.3 · Minor Revisions

Thank you for the revisions. I appreciate the description of your screening process, especially the methods used to address potential discrepancies. Please state clearly in the manuscript whether any discrepancies occurred during the screening process.

---

## Round 0.4 · accepted · Accept

Thank you for your revised submission. I am satisfied that you have addressed the remaining concerns of the reviewers, and am happy to accept your paper for publication.